# Association between Achievement of Estimated Average Glucose Level and 6-Month Neurologic Outcome in Comatose Cardiac Arrest Survivors: A Propensity Score-Matched Analysis

**DOI:** 10.3390/jcm8091480

**Published:** 2019-09-18

**Authors:** Yong Hun Jung, Byung Kook Lee, Kyung Woon Jeung, Dong Hun Lee, Hyoung Youn Lee, Yong Soo Cho, Chun Song Youn, Jung Soo Park, Yong II Min

**Affiliations:** 1Department of Emergency Medicine, Chonnam National University, Chonnam National University Hospital, 160, Beakseo-ro, Dong-gu, Gwangju 61469, Korea; xnxn77@hanmail.net (Y.H.J.); neoneti@hanmail.net (K.W.J.); ggodhkekf@hanmail.net (D.H.L.); apostle09@naver.com (H.Y.L.); semi-moon@hanmail.net (Y.S.C.); minyi46@hanmail.net (Y.I.M.); 2Department of Emergency Medicine, College of Medicine, The Catholic University of Korea, Seoul 06591, Korea; ycs1005@catholic.ac.kr; 3Department of Emergency Medicine, College of Medicine, Chungnam National University, Daejeon 35015, Korea; Chopin-park@hanmail.net

**Keywords:** heart arrest, prognosis, glucose, glycated hemoglobin A

## Abstract

We investigated whether achieving estimated average glucose (EAG) levels versus achieving standard glucose levels (180 mg/dL) was associated with neurologic outcome in cardiac arrest survivors. This single-center retrospective observational study included adult comatose cardiac arrest survivors undergoing therapeutic hypothermia (TH) from September 2011 to December 2017. EAG level was calculated using HbA1c obtained after the return of spontaneous circulation (ROSC), and the mean glucose level during TH was calculated. We designated patients to the EAG or standard glucose group according to whether the mean blood glucose level was closer to the EAG level or 180 mg/dL. Patients in the EAG and standard groups were propensity score- matched. The primary outcome was the 6-month neurologic outcome. The secondary outcomes were hypoglycemia (≤70 mg/dL) and serum neuron-specific enolase (NSE) at 48 h after ROSC. Of 384 included patients, 137 (35.7%) had a favorable neurologic outcome. The EAG group had a higher favorable neurologic outcome (104/248 versus 33/136), higher incidence of hypoglycemia (46/248 versus 11/136), and lower NSE level. After propensity score matching, both groups had similar favorable neurologic outcomes (24/93 versus 27/93) and NSE levels; the EAG group had a higher incidence of hypoglycemia (21/93 versus 6/93). Achieving EAG levels was associated with hypoglycemia but not neurologic outcome or serum NSE level.

## 1. Introduction

Glucose control is an important therapeutic strategy in comatose cardiac arrest survivors [1,2]. Hyperglycemia after cardiac arrest is one of the common adverse events, and it is associated with increased mortality or unfavorable neurologic outcome [3,4,5]. Counter-regulatory hormones such as catecholamines cause stress-induced hyperglycemia in critical illness [6]. However, the optimal target glucose level for post-cardiac arrest care remains unknown. A randomized controlled trial proving that conventional glucose control results in lower mortality than strict glucose control in critically ill patients provided a target glucose level of less than 180 mg/dL [7]. Another randomized trial involving ventricular fibrillation out-of-hospital cardiac arrest (OHCA) revealed that there is no difference in mortality between strict (72–108 mg/dL) and moderate (108–144 mg/dL) glucose control [8]. Although there is a lack of evidence, the European Resuscitation Council recommends adjusting blood glucose levels to less than 180 mg/dL after the return of spontaneous circulation (ROSC) in these and other patients with critical illness [1,7]. 

Glycated hemoglobin (HbA1c) is not only used as a diagnostic measure for diabetes; it also reflects the average glycemic status during the previous 4 to 12 weeks [9,10,11]. The estimated average glucose (EAG) can be calculated from the HbA1c for practical convenience [12]; one study suggested that the EAG can be a reference value for managing blood glucose levels among in-hospital cardiac arrest survivors with diabetes [13]. We hypothesized that EAG calculated using HbA1c can be used as a target glucose level, rather than 180 mg/dL because HbA1c can provide information regarding the glycemic status right before development of cardiac arrest. Therefore, we investigated the difference in outcomes between patients achieving standard glucose and EAG levels, using propensity score matching.

## 2. Patients and Methods

### 2.1. Study Design and Patients

We performed a retrospective observational study using prospectively collected data of adult comatose cardiac arrest survivors undergoing therapeutic hypothermia (TH) at Chonnam National University Hospital in Gwangju, Korea, from September 2011 to December 2017. The Institutional Review Board of Chonnam National University Hospital approved this study (CNUH-2018-207).

We included adult (≥18 years) cardiac arrest survivors who underwent TH in whom HbA1c was measured after ROSC. We excluded patients who had interrupted TH owing to transfer or death, had no data on HbA1c, lacked glucose data during TH, those treated with a target temperature other than 33 °C and/or a target duration other than 24 h (e.g., 36 °C and/or 72 h), and those who were supported with extracorporeal membrane oxygenation during post-cardiac arrest care. We defined the EAG and standard groups according to whether the mean blood glucose level during TH was closer to the EAG level or 180 mg/dL. 

### 2.2. Therapeutic Hypothermia and Measurement and Glucose Control

Comatose cardiac arrest survivors underwent TH in accordance with the guidelines. A target temperature of 33 °C was maintained for 24 h using a feedback-controlled device. Upon completion of the TH maintenance phase, patients were rewarmed at a rate of 0.25–0.5 °C/hour. All patients received continuous intravenous midazolam and remifentanil (or fentanyl). Other aspects of patient management were at the discretion of the treating physician. 

Nurses maintained target blood glucose levels within the range 80–200 mg/dL using intravenous insulin or glucose, in accordance with a written protocol (Table 1). We avoided glucose-containing solutions whenever possible during TH unless hypoglycemia (≤70 mg/dL) was identified. We measured blood glucose using arterial blood obtained from arterial catheters with Accu-check (Roche/Hitachi, Basel, Switzerland), at least every 4 h. If hypoglycemia or severe hyperglycemia (>350 mg/dL) was documented, we performed additional glucose measurement after infusion of glucose or insulin, according to the protocol.

### 2.3. Data Collection and Primary Outcome

We obtained the following data from the hospital records: age, sex, pre-existing illness, body mass index, location of cardiac arrest, presence of a witness on collapse, bystander cardiopulmonary resuscitation (CPR), first monitored rhythm, etiology of cardiac arrest, adrenaline dose used during CPR, time to ROSC, hemoglobin, serum lactate, glucose, and HbA1c after ROSC, partial pressure of oxygen (PaO_2_) and carbon dioxide (PaCO_2_) after ROSC, Glasgow Coma Scale (GCS) after ROSC, sequential organ failure assessment (SOFA) score within the first 24 h after admission [14], time from ROSC to TH, induction duration, rewarming duration, and neuron-specific enolase (NSE) at 48 h after cardiac arrest. 

We calculated EAG using the following equation: EAG = 28.7 × HbA1c − 46.7 (mg/dL) [8]. We defined hypoglycemia as glucose level below 70 mg/dL. Glucose measurements throughout TH were recorded, and mean glucose levels during TH were calculated. Mean glucose was divided into tertiles, and HbA1c was divided into three groups: non-diabetic (4.0–5.6%), pre-diabetic (5.7–6.4%), and diabetic (6.5–11.8%) [15]. 

A board-certified emergency physician assessed the Cerebral Performance Category (CPC) scale via phone interview of patients or patient representatives and recorded as follows: CPC 1, good performance; CPC 2, moderate disability; CPC 3, severe disability; CPC 4, coma or vegetative state; or CPC 5, brain death or death [16]. The primary outcome was an unfavorable neurologic outcome, defined as CPC 3–5, and secondary outcomes were hypoglycemia and serum NSE at 48 h after cardiac arrest.

### 2.4. Statistical Analysis

We described categorical variables as frequencies and percentages and continuous variables as median values with interquartile range (IQR) or mean with standard deviation (SD). We compared categorical variables between groups using χ^2^ or Fisher’s exact tests, as appropriate. We compared continuous variables between groups using the Mann–Whitney *U* test or independent *t*-test according to a normality test. 

To reduce selection bias and potential confounding factors in this observational study, we performed a propensity score matching. We calculated a propensity score using a multivariate logistic regression model based on variables with *p* < 0.2 between the EAG and standard groups. We performed propensity score matching in a one-to-one fashion between the EAG and standard group using calipers with a width equal to 0.2 of the SD of the logit of the propensity score. We assessed the differences in distributions of variables by absolute standardized differences. We considered an absolute standardized difference ≥10% indicative of a significant difference between the EAG and standard groups. We used McNemar’s test for categorical variables and the Wilcoxon signed-rank test or paired *t*-test for continuous variables in the matched cohort. 

We used univariate and multivariate binary logistic regression analyses to assess the association between glucose groups and neurologic outcome. All variables with *p* < 0.2 in univariate comparisons between neurologic outcome groups (Appendix A) were included in the multivariate regression model. We assessed the collinearity between variables before modeling. We used a backward stepwise approach, sequentially eliminating variables with a threshold of *p* > 0.10 to establish a final adjusted regression model. Goodness-of-fit of the final model was evaluated using the Hosmer–Lemeshow test. We included covariates which were independent predictors of neurologic outcome in the final model (Appendix A). To identify the interaction between HbA1c and mean glucose, we calculated observed and adjusted neurologic outcomes in the groups created according to the mean glucose tertiles and the three subgroups of HbA1c. We defined adjusted neurologic outcome as the observed unfavorable neurologic outcome in each group divided by the predicted outcome calculated from the final multivariate logistic regression model except for HbA1c. Univariate association between the glucose group and outcome and the final model adjusted for the multivariate association were tested in the subgroups of HbA1c. In the propensity score-matched cohort, we used conditional logistic regression model matched-pair data. We present the logistic regression analysis results as odds ratio (OR) and 95% confidence interval (CI). We set the statistical significance as a two-sided *p*-value < 0.05. We analyzed data using Stata version 13.1 (StataCorp, College Station, TX, USA) and IBM SPSS for Windows version 18.0 (IBM Corp., Armonk, NY, USA). 

## 3. Results

We treated 635 adult comatose cardiac arrest survivors with TH during the study period; 384 of these patients were included in the present study (Figure 1). EAG was 117 mg/dL (IQR, 105–134 mg/dL), and mean glucose level during TH was 137 mg/dL (IQR, 118–164 mg/dL). We categorized patients into the EAG (*n* = 248, 64.6%) or standard (*n* = 136, 35.4%) glucose groups. Mean glucose levels during TH were 124 mg/dL (IQR, 112–138 mg/dL) in the EAG group and 172 mg/dL (IQR, 156–188 mg/dL) in the standard group (*p* < 0.001). Figure 2 shows the glucose levels according to EAG or standard group throughout TH. 

Table 2 shows baseline characteristics stratified according to the EAG or standard group in the entire cohort. Patients in the EAG group were younger, had a higher incidence of diabetes, were more likely to have an OHCA and shockable rhythm, and required lower doses of adrenaline. The EAG group also had lower HbA1c, glucose, and SOFA and higher hemoglobin after ROSC. The EAG group achieved the target temperature after the initiation of TH later than the standard group.

### 3.1. Matched Cohort

After propensity score matching, each group included 93 patients (Figure 1 and Table 3). Mean glucose levels during TH were 126 mg/dL (IQR, 113–138 mg/dL) in the EAG group and 172 mg/dL (IQR, 157–191 mg/dL) in the standard glucose group (*p* < 0.001). No variables were significantly different between the EAG and standard groups after propensity score matching, and absolute standard differences in the matched cohort were improved (Table 3).

### 3.2. Outcomes

Unfavorable neurologic outcome at 6 months after cardiac arrest occurred in 247 (64.3%) patients, and hypoglycemia developed in 57 (14.8%), among the entire cohort. Figure 3 shows the associations between outcomes and the EAG or standard group. The EAG group was associated with favorable neurologic outcome (*p* < 0.001), higher incidence of hypoglycemia (*p* = 0.006), and lower serum NSE levels at 48 h after cardiac arrest (*p* = 0.001), in the entire cohort (Figure 3A). However, the EAG group versus the standard group was not independently associated with unfavorable neurologic outcome at 6 months after cardiac arrest (OR, 0.896; 95% CI, 0.455–1.764), after adjustment for the entire cohort (Table 4). The observed and adjusted unfavorable neurologic outcome according to tertiles of mean glucose and subgroups of HbA1c is shown in Figure 4. EAG group was associated with increased hypoglycemia only in subgroup of non-diabetic and pre-diabetic patients (EAG, 31/210 versus standard 6/93; *p* = 0.010), whereas EAG group was not associated with hypoglycemia in diabetic subgroup (EAG, 9/38 versus standard 5/43; *p* = 0.152).

Unfavorable neurologic outcome occurred in 135 (72.6%), and hypoglycemia developed in 27 (14.5%) patients in the propensity score-matched cohort. The EAG group was not associated with neurologic outcome (*p* = 0.728) and serum NSE level (*p* = 0.501) at 48 h after cardiac arrest but was associated with a higher incidence of hypoglycemia (*p* = 0.004; Figure 3B). The EAG group versus the standard group showed no independent association with unfavorable neurologic outcome at 6 months after cardiac arrest (OR, 0.540; 95% CI, 0.143–2.034), after adjustment for the propensity score-matched cohort (Table 4). EAG group was associated with increased hypoglycemia only in subgroup of non-diabetic and pre-diabetic patients (EAG, 18/78 versus standard 3/72; *p* = 0.001), whereas it was not associated with hypoglycemia in diabetic subgroup (EAG, 3/15 versus standard 3/21; *p* = 0.677).

## 4. Discussion

In this propensity score-matched study, we compared the achievement of EAG levels with achievement of standard glucose levels in comatose cardiac arrest survivors undergoing TH. We found no significant difference in 6-month neurologic outcome and serum NSE levels. However, the EAG group had a higher incidence of hypoglycemia during TH than the group achieving standard glucose levels. 

There is a clear association between hyperglycemia after ROSC and worse clinical outcome in comatose survivors of cardiac arrest [17,18,19,20]. However, the causation between hyperglycemia and neurologic outcome in comatose cardiac arrest survivors has not been clarified, and the optimal glucose target range has not been identified. Experts suggest moderate glucose control of blood glucose levels below 180 mg/dL for comatose survivors of cardiac arrest, based on the NICE-SUGAR trial [1,7]. The NICE-SUGAR trial demonstrated that intensive glucose control aiming at blood glucose levels of 81–108 mg/dL increased 90-day mortality by 14% in critically ill patients, as compared with conventional control of blood glucose below 180 mg/dL [7]. However, similar to a prior randomized trial by Oksanen et al. demonstrating that both strict and moderate glucose control had comparable 30-day mortality in 90 ventricular fibrillation OHCA survivors [8], the group achieving blood glucose control was not independently associated with 6-month neurologic outcome in the present study, irrespective of whether in the entire cohort or the propensity score-matched cohort. Considering the target glucose values, the NICE-SUGAR trial compared the intensive (81–108 mg/dL) with the conventional (below 180 mg/dL) method, and a study by Oksanen et al. compared intensive (72–108 mg/dL) and moderate (108–144 mg/dL) methods [7,8]. The EAG and standard groups in the present study were close to the moderate glucose control and conventional glucose control groups, respectively. Therefore, according to the target glucose value alone, the NICE-SUGAR trial seems to have a high probability of different outcomes between groups. However, in a comparison of the achieved glucose level in the present study, the EAG group (median, 124 mg/dL; IQR, 112–138 mg/dL) had mean glucose levels that were comparable to those of the intensive glucose control group (115 ± 18 mg/dL) in the NICE-SUGAR trial, and the standard group (median, 172 mg/dL; IQR, 156–188 mg/dL) in the present study had higher mean glucose levels than the conventional glucose control range (144 ± 23 mg/dL) in the NICE-SUGAR trial [7]. Whereas the difference in mean glucose levels between the EAG and standard groups seems to be higher than the difference between the intensive and conventional glucose control groups in the NICE-SUGAR trial, neurologic outcome was not different in the present study. Although it may be inappropriate to make a direct comparison because the research participants are different, the above might imply that the glucose control strategy is more critical to clinical outcome rather than the actual glucose level. The intensive and conventional groups in the NICE-SUGAR trial were managed with different protocols [7] whereas the EAG and standard groups were managed within the same target range of 80–200 mg/dL in the present study. 

Although the multivariate models and propensity-score matched cohort failed to show any significant association between the glucose group and neurologic outcome, the final model adjusted for the interaction of HbA1c and mean glucose showed that the association between hyperglycemia and the neurologic outcome might differ according to the premorbid glycemic status. Previous studies demonstrated that liberal (180–252 mg/dL) glucose control was not associated with increased mortality or adverse events, whereas liberal glucose control was associated with reduced hypoglycemia and glucose variability than conventional (108–180 mg/dL) glucose control in critically ill patients with diabetes or patients with chronic hyperglycemia (HbA1c ≥ 7%) [21,22,23]. That previous research supports that HbA1c based glucose control might be better to avoid potential risk of adverse events in cardiac arrest survivors with chronic hyperglycemia or poorly controlled diabetes. 

Intensive glucose control leading to more hypoglycemic episodes and hypoglycemia is associated with increased mortality in critically ill patients [7,24]. Oksanen et al. demonstrated that strict glucose control leads to the higher incidence of hypoglycemia in OHCA survivors and suggested that moderate glucose control is acceptable owing to the potentially detrimental effect of hypoglycemia arising from strict glucose control [8]. Overall, 57 of our patients (14.8%) experienced at least one hypoglycemic episode during TH, which was higher than the 7% of patients with hypoglycemic episodes in the TTM trial [20]; this is believed to be associated with a lack of enteral or parenteral feeding during TH, similar to prior work reporting 20% of OHCA survivors undergoing TH who experienced hypoglycemic episodes [18]. The EAG group had significantly higher hypoglycemia for both the entire and propensity score-matched cohorts in our study. Although it is not obvious whether TH and/or treatment including insulin and glucose-free fluid cause hypoglycemia, it seems clear that the EAG group had a higher risk for the development of hypoglycemia since it had a lower mean glucose level than the standard group. However, diabetic subgroup with the HbA1c ≥ 6.5% was not associated with hypoglycemia, unlike the non-diabetic and pre-diabetic subgroups in the present study. Previous studies demonstrated that liberal glucose control reduced hypoglycemia in critically ill patients with HbA1c ≥ 7% [22,23]. Thus, even if the association between hypoglycemia and unfavorable neurologic outcome is unclear in the present study, it might be acceptable to recommend the EAG based glucose control in cardiac arrest survivors with chronic hyperglycemic status to avoid potential harm owing to hypoglycemia. 

NSE is often used as an outcome measure, with or without clinical outcome, to estimate the effectiveness of an intervention in cardiac arrest survivors [8,25,26]. Although serial measurement of NSE provides more accurate prognostic information, a single measurement of NSE at 48 or 72 h has also shown high prognostic performance in OHCA survivors [27]. Oksanen et al. demonstrated that the NSE at 24 h and 48 h after ROSC was not different between glucose control groups [8]. Similar to prior work, we used NSE at 48 h after ROSC and the NSE level was not different between the EAG and standard groups in the propensity score-matched cohort.

There are several limitations in the present study. Owing to the observational nature of this study, we can only report an association but not causation. About 40% of patients undergoing TH were excluded, which might lead to selection bias. We managed blood glucose in both groups with the same protocol, aiming for a blood glucose level between 80 and 200 mg/dL, which differs from the European Resuscitation Council guideline [1]; in addition, the groups were designated according to the actual glucose level achieved, not the target glucose level of EAG or 180 mg/dL. Therefore, interpretation of the findings should be made with caution. The mean glucose level depends on the period included in the calculation. The longer the period included in the calculation, the greater the likelihood of being classified in the EAG group because blood glucose levels decrease over time throughout TH. We calculated mean blood glucose throughout TH to define each group. Although the EAG group seemed to be closer to 180 mg/dL at the initiation of TH, the standard group became closer to 180 mg/dL after 4 h of maintenance and the SD of mean glucose level in the EAG group continued to be under 180 mg/dL after 8 h of maintenance. The difference in mean glucose level between the standard and EAG groups seems to be larger than the difference in mean glucose level between the standard group and 180 mg/dL (Figure 2). Nevertheless, blood glucose levels overlapped between the two groups throughout TH (Figure 2), which suggests ambiguity in defining the EAG or standard groups. We consider this an inevitable limitation derived from the definition of groups based on the achieved glucose level and not the target glucose level. We performed propensity score-matching to balance the confounding factors; however, a few factors still had considerable differences. Therefore, we used multivariate logistic regression analysis again in the propensity score-matched cohort, to control these covariates.

## 5. Conclusions

Achievement of EAG levels during TH in comatose cardiac arrest survivors was not associated with 6-month neurologic outcome or serum NSE in our propensity score-matched cohort, whereas it was associated with a high prevalence of hypoglycemia. However, the associations seem to be invalid in diabetic subgroup. Therefore, further randomized clinical trials are warranted to identify the optimal glucose range based on the chronic glycemic status after ROSC in cardiac arrest survivors. 

## Figures and Tables

**Figure 1 jcm-08-01480-f001:**
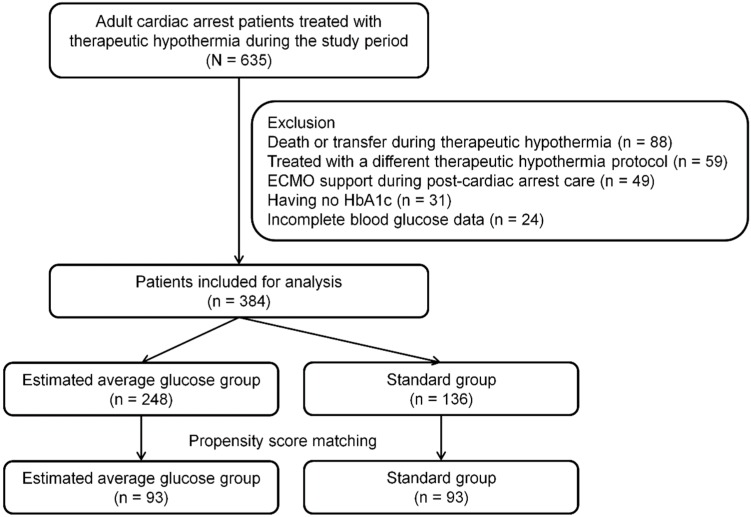
Diagram showing the number of included patients. ECMO, extracorporeal membrane oxygenation.

**Figure 2 jcm-08-01480-f002:**
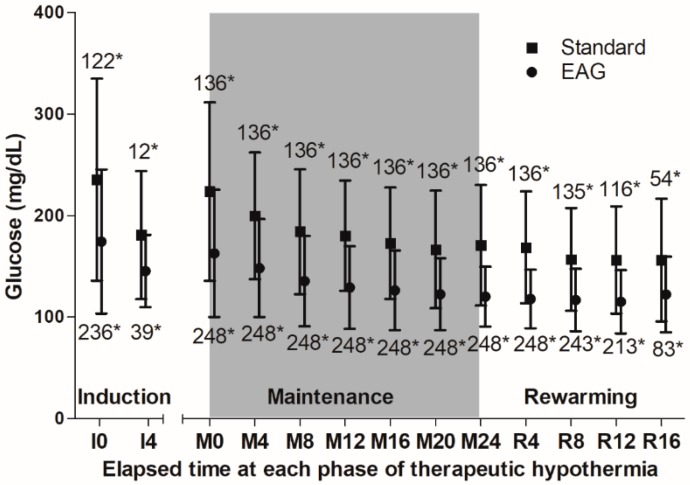
Serum glucose level during therapeutic hypothermia according to estimated average glucose or standard glucose groups. I, induction phase; M, maintenance phase; R, rewarming phase. *, Number of patients included in the analysis.

**Figure 3 jcm-08-01480-f003:**
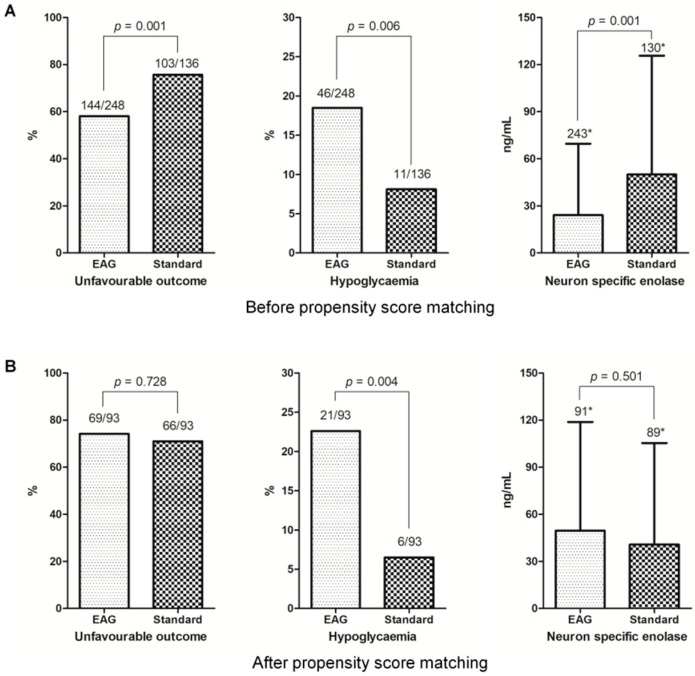
Outcomes according to estimated average glucose (EAG) or standard groups. (**A**) The EAG group had significantly lower unfavorable neurologic outcomes (*p* = 0.001), lower serum neuron-specific enolase (*p* = 0.001), and higher incidence of hypoglycemia (*p* = 0.006) in the entire cohort. (**B**) The EAG group showed similar neurologic outcomes (*p* = 0.728) to the standard group and had a neuron-specific enolase level (*p* = 0.501) comparable to the standard group, and higher incidence of hypoglycemia (*p* = 0.004). * Number of patients included in the analysis.

**Figure 4 jcm-08-01480-f004:**
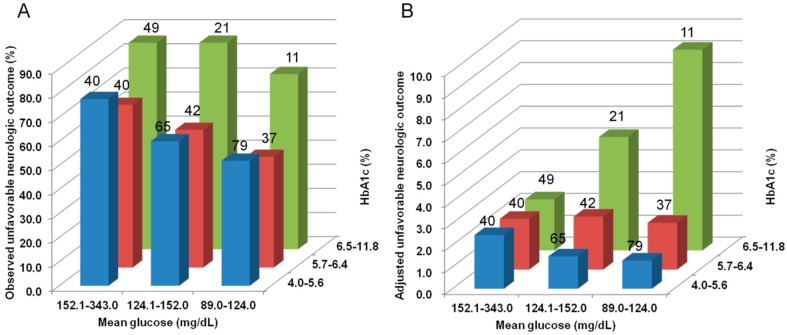
Observed and adjusted neurologic outcomes. (**A**) The observed unfavorable neurologic outcome increased with an increase in the HbA1c and mean glucose. (**B**) Adjusted unfavorable neurologic outcome increased in the non-diabetic (4.0%–5.6%) and pre-diabetic (5.7%–6.4%) groups whereas adjusted unfavorable neurologic outcome decreased in the diabetic (6.5%–11.8%) group with an increase in the mean glucose. Adjusted unfavorable neurologic outcome increased in the first and second mean glucose tertiles with an increase in the HbA1c. However, the adjusted unfavorable neurologic outcome in the third mean glucose tertile was constant irrespective of HbA1c.

**Table 1 jcm-08-01480-t001:** Glucose control protocol.

1. Target glucose level: 80–200 mg/dL	
2. Measure blood glucose every four hours in blood samples obtained from an arterial catheter using the Accu-check, except if hypoglycemia (<70 mg/dL) or hyperglycemia (>350 mg/dL) are observed.	
3. Intervention for glucose control	
**Glucose Level**	**Intervention**
<60 mg/dL	Administer 50 mL of D20W intravenously and recheck glucose in 30 min. If glucose remains <60 mg/dL, repeat intravenous administration of 50 mL of D20W every 30 min until glucose is >60 mg/dL
60–79 mg/dL	Do not give an insulin injection. Recheck blood glucose every hour. If glucose remains <70 mg/dL, administer 25 mL of D20W intravenously every hour until glucose is >70 mg/dL
80–200 mg/dL	Do not give an insulin injection. Recheck blood glucose every 4 h
201–250 mg/dL	Administer 2 U of regular insulin intravenously. Recheck blood glucose every 4 h
251–350 mg/dL	Administer 4 U of regular insulin intravenously. Recheck blood glucose every 4 h
>350 mg/dL	Administer 6 U of regular insulin intravenously. Recheck blood glucose in 1 h

**Table 2 jcm-08-01480-t002:** Demographic, cardiac arrest, and clinical characteristics between EAG and standard groups.

	Total (*n* = 384)	EAG (*n* = 248)	Standard (*n* = 136)	*p*	ASD
Demographic characteristics					
Age (years), median (IQR)	61.0 (50.0–70.0)	59.0 (47.0–68.0)	63.5 (53.3–72.0)	0.002	0.326
Male, *n* (%)	255 (66.4)	170 (68.5)	85 (62.5)	0.230	0.112
Pre-existing illness, *n* (%)					
Coronary artery disease	65 (16.9)	36 (14.5)	29 (21.3)	0.089	0.170
Congestive heart failure	37 (9.6)	24 (9.7)	13 (9.6)	0.970	0.004
Hypertension	167 (43.5)	99 (39.9)	68 (50.0)	0.057	0.176
Diabetes	115 (29.9)	54 (21.8)	61 (44.9)	<0.001	0.511
Pulmonary disease	18 (4.7)	10 (4.0)	8 (5.9)	0.412	0.083
Renal impairment	48 (12.5)	30 (12.1)	18 (13.2)	0.747	0.032
Cerebrovascular accident	27 (7.0)	16 (6.5)	11 (8.1)	0.549	0.060
Hepatic disease	6 (1.6)	4 (1.6)	2 (1.5)	1.000	0.011
Body mass index (kg/m^2^), median (IQR)	22.9 (21.3–25.2)	23.1 (21.3–25.4)	22.8 (21.1–24.8)	0.337	0.156
Cardiac arrest characteristics					
OHCA, *n* (%)	319 (83.1)	216 (87.1)	103 (75.7)	0.005	0.215
Witnessed, *n* (%)	286 (74.5)	182 (73.4)	104 (76.5)	0.507	0.054
Bystander CPR, *n* (%)	240 (62.5)	156 (62.9)	84 (61.8)	0.826	0.018
Shockable rhythm, *n* (%)	130 (33.9)	94 (37.9)	36 (26.5)	0.024	0.194
Cardiac etiology, *n* (%)	214 (55.7)	141 (56.9)	73 (53.7)	0.549	0.050
Adrenaline (mg), median (IQR)	2.0 (1.0–4.0), 381 *	2.0 (0.0–4.0), 245 *	3.0 (2.0–5.0)	<0.001	0.311
Time to ROSC (min), median (IQR)	27.0 (15.0–40.0)	25.0 (15.0–37.0)	30.0 (16.0–42.0)	0.045	0.266
Clinical characteristics					
HbA1c (%), median (IQR)	5.7 (5.3–6.3)	5.6 (5.3–6.1)	5.9 (5.4–7.0)	<0.001	0.494
Hemoglobin (mg/dL), median (IQR)	13.0 (10.9–14.8)	13.3 (11.2–14.9)	12.1 (10.6–14.5)	0.023	0.220
Lactate (mmol/L), median (IQR)	7.4 (4.2–10.3)	6.8 (4.2–9.7)	8.1 (4.3–10.9)	0.075	0.174
Glucose (mg/dL), median (IQR)	230 (169–301)	214 (159–274)	281 (201–337)	<0.001	0.525
PaO_2_ (mmHg), median (IQR)	137 (85–220)	137 (87–220)	139 (81–223)	0.845	0.034
PaCO_2_ (mmHg), median (IQR)	38.0 (30.9–47.0)	38.0 (31.0–46.0)	38.4 (30.1–48.8)	0.516	0.101
GCS, median (IQR)	3 (3–3)	3 (3–4)	3 (3–3)	0.083	0.224
SOFA score, median (IQR)	9 (7–12)	8 (6–11)	10 (7–13)	<0.001	0.372
Time from ROSC to TH (min), median (IQR)	235 (175–310)	240 (180–315)	231 (163–302)	0.221	0.120
Induction duration (h), median (IQR)	2.3 (1.3–3.3)	2.5 (1.5–3.5)	2.0 (1.0–2.8)	0.001	0.360
Rewarming duration (h), median (IQR)	13.0 (12.0–16.0)	13.0 (12.0–16.0)	14.0 (12.0–16.0)	0.244	0.155

EAG, estimated average glucose; ASD, absolute standardized difference; IQR interquartile range; OHCA, out-of-hospital cardiac arrest; CPR, cardiopulmonary resuscitation; ROSC, restoration of spontaneous circulation; GCS, Glasgow Coma Scale; SOFA, sequential organ failure assessment; TH therapeutic hypothermia; HbA1c, glycated hemoglobin; PaO_2_, partial pressure of oxygen; PaCO_2_, partial pressure of carbon dioxide. *, Number of patients included in the analysis.

**Table 3 jcm-08-01480-t003:** Demographic, cardiac arrest, and clinical characteristics between EAG and standard glucose groups in the propensity score-matched cohort.

	Total (*n* = 186)	EAG (*n* = 93)	Standard (*n* = 93)	*p*	ASD
Demographic characteristics					
Age (years), median (IQR)	62.0 (51.0–70.0)	63.0 (51.0–70.0)	60.0 (50.5–69.5)	0.442	0.101
Male, *n* (%)	122 (65.6)	61 (65.6)	61 (65.6)	1.000	0.000
Pre-existing illness, *n* (%)					
Coronary artery disease	29 (15.6)	10 (10.8)	19 (20.4)	0.078	0.283
Congestive heart failure	17 (9.1)	7 (7.5)	10 (10.8)	0.629	0.114
Hypertension	82 (44.1)	44 (47.3)	38 (40.9)	0.488	0.126
Diabetes	52 (28.0)	26 (28.0)	26 (28.0)	1.000	0.000
Pulmonary disease	11 (5.9)	5 (5.4)	6 (6.5)	1.000	0.046
Renal impairment	17 (9.1)	10 (10.8)	7 (7.5)	0.629	0.110
Cerebrovascular accident	15 (8.1)	8 (8.6)	7 (7.5)	1.000	0.039
Hepatic disease	1 (0.5)	0 (0.0)	1 (1.1)	NA	0.148
Body mass index (kg/m^2^), median (IQR)	23.1 (21.3–25.4)	23.1 (21.2–25.4)	23.2 (21.4–25.2)	0.948	0.023
Cardiac arrest characteristics					
OHCA, *n* (%)	153 (82.3)	76 (81.7)	77 (82.8)	1.000	0.028
Witnessed, *n* (%)	134 (72.0)	66 (71.0)	68 (73.1)	0.864	0.048
Bystander CPR, *n* (%)	118 (63.4)	60 (64.5)	58 (62.4)	0.880	0.044
Shockable rhythm, *n* (%)	54 (29.0)	26 (28.0)	28 (30.1)	0.871	0.048
Cardiac etiology, *n* (%)	95 (51.1)	44 (47.3)	51 (54.8)	0.360	0.157
Adrenaline (mg), median (IQR)	2.0 (1.0–5.0), 185 *	3.0 (1.0–5.0), 92 *	2.0 (1.0–5.0)	0.975	0.041
Time to ROSC (min), median (IQR)	28.0 (15.8–40.0)	29.0 (15.0–40.0)	28.0 (16.0–39.5)	0.523	0.082
Clinical characteristics					
HbA1c (%), median (IQR)	5.6 (5.3–6.2)	5.6 (5.3–6.1)	5.7 (5.4–6.4)	0.053	0.143
Hemoglobin (mg/dL), median (IQR)	12.6 ± 2.6	12.4 ± 2.5	12.8 ± 2.6	0.366	0.135
Lactate (mmol/L), median (IQR)	7.8 (4.2–10.3)	7.0 (4.2–10.3)	8.0 (4.3–10.4)	0.866	0.123
Glucose (mg/dL), median (IQR)	235 ± 84	232 ± 84	239 ± 85	0.558	0.081
PaO_2_ (mmHg), median (IQR)	140 (92–232)	134 (96–229)	145 (86–245)	0.805	0.014
PaCO_2_ (mmHg), median (IQR)	36.5 (29.1–47.0)	35.7 (29.0–47.0)	38.0 (30.0–47.2)	0.708	0.041
Glasgow Coma Scale, median (IQR)	3 (3–3)	3 (3–3)	3 (3–3)	0.906	0.046
SOFA score, median (IQR)	9 (7–12)	10 (7–12)	9 (7–12)	0.769	0.066
Time from ROSC to TH (min), median (IQR)	240 (185–309)	260 (199–313)	234 (165–301)	0.178	0.191
Induction duration (h), median (IQR)	2.0 (1.0–3.0)	2.3 (1.0–3.0)	2.0 (1.1–2.8)	0.818	0.089
Rewarming duration (h), median (IQR)	14.0 (12.0–16.0)	13.0 (12.0–16.0)	14.0 (12.0–16.0)	0.948	0.011

EAG, estimated average glucose; ASD, absolute standardized difference; IQR interquartile range; NA, not applicable; OHCA, out-of-hospital cardiac arrest; CPR, cardiopulmonary resuscitation; ROSC, restoration of spontaneous circulation; SOFA, sequential organ failure assessment; TH therapeutic hypothermia; HbA1c, glycated hemoglobin; PaO_2_, partial pressure of oxygen; PaCO_2_, partial pressure of carbon dioxide. * Number of patients included in the analysis.

**Table 4 jcm-08-01480-t004:** Odds ratios of estimated average glucose versus standard group for unfavorable neurologic outcome.

Model	Number	Odds Ratio (95% Confidence Interval)	*p*
Crude	384	0.444 (0.278–0.707)	0.001
Non-diabetic and pre-diabetic	303	0.477 (0.282–0.804)	0.006
Diabetic	81	0.718 (0.218–2.361)	0.586
Adjusted	384	0.896 (0.455–1.764)	0.750
Non-diabetic and pre-diabetic	303	0.974 (0.469–2.022)	0.943
Diabetic	81	0.314 (0.037–2.697)	0.291
Crude in matched cohort	186	1.176 (0.617–2.242)	0.622
Non-diabetic and pre-diabetic	150	1.273 (0.635–2.551)	0.497
Diabetic	36	1.083 (0.158–7.435)	0.935
Adjusted in matched cohort	186	0.540 (0.143–2.034)	0.362
Non-diabetic and pre-diabetic	150	1.157 (0.448–2.990)	0.763
Diabetic	36	3.827 (0.111–131.563)	0.457

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
