# Peer review of "Association between Achievement of Estimated Average Glucose Level and 6-Month Neurologic Outcome in Comatose Cardiac Arrest Survivors: A Propensity Score-Matched Analysis"

_jcm, 2019, doi:10.3390/jcm8091480_

Round 1

Reviewer 1 Report

Comments:

It’s not clear how the glucose is maintained. Continuously of just for a short period of time just before induction of hypothermia? “Nurses maintained target blood glucose levels within the range 80–200 mg/dL using intravenous insulin or glucose, “ , if such levels of glucose are maintained continuously, why  hypoglycemia (≤ 70 mg/dL) occurs? “If hypoglycemia or severe hypoglycemia was documented, we performed additional glucose measurement after infusion of glucose or insulin, according to the protocol” , it needs to be clarified. What is “severe hypoglycemia” ? Why insulin infusion is needed? What’s the reason for hypoglycemia? Is it due to TH or is it part of post-resuscitation syndrome? The EAG group has lower basal level of glucose compared to the standard group, If TH induces hypoglycemia, then it is not surprising that the EAG has higher incidence of hypoglycemia. The 6-month neurological outcome was evaluated by a “phone interview” rather than a neurological exam, the accuracy of the response may need to be confirmed. The conclusion is very vague. The EAG group and standard group had similar neurological outcomes, though the EAG groups had higher incidence of hypothermia. In 3.1, the first two paragraphs are exactly the same.

Author Response

We appreciate you for kind review. They were very helpful in improving our manuscript and included very useful points that we had not previously recognized. After due consideration, the manuscript was revised as described below.

1. It’s not clear how the glucose is maintained. Continuously of just for a short period of time just before induction of hypothermia? “Nurses maintained target blood glucose levels within the range 80–200 mg/dL using intravenous insulin or glucose, “, if such levels of glucose are maintained continuously, why hypoglycemia (≤ 70 mg/dL) occurs? If hypoglycemia or severe hypoglycemia was documented, we performed additional glucose measurement after infusion of glucose or insulin, according to the protocol”, it needs to be clarified. What is “severe hypoglycemia”? Why insulin infusion is needed?

Answer) We monitored the blood glucose at intervals that were at least 4 hours apart and not continuously. However, if the patient had hypoglycemia or severe hyperglycemia (> 350 mg/dL), blood glucose was measured frequently. Most patients had hyperglycemia immediately after ROSC. Therefore, blood glucose was measured frequently immediately after ROSC. When the patient blood glucose reached 80–200 mg/dL, measurements were performed every 4 hours.

We added the following Glucose control protocol as Table 1 to clarify this.

Table 1. Glucose control protocol

1. Target glucose level: 80–200 mg/dL

2. Measure blood glucose every four hours in blood samples obtained from an arterial catheter using the Accu-check, except if hypoglycemia (< 70 mg/dL) or hyperglycemia (> 350 mg/dL) are observed.

3. Intervention for glucose control

Glucose level

Intervention

< 60 mg/dL

Administer 50 mL of D20W intravenously and recheck glucose in 30 min. If glucose remains < 60 mg/dL, repeat intravenous administration of 50 mL of D20W every 30 min until glucose is > 60 mg/dL

60 – 79 mg/dL

Do not give insulin injection. Recheck blood glucose every hour. If glucose remains < 70 mg/dL, administer 25 mL of D20W intravenously every hour until glucose is > 70 mg/dL

80 – 200 mg/dL

Do not give an insulin injection. Recheck blood glucose every 4 hours

201 – 250 mg/dL

Administer 2 U of regular insulin intravenously. Recheck blood glucose every 4 hours

251 – 350 mg/dL

Administer 4 U of regular insulin intravenously. Recheck blood glucose every 4 hours

> 350 mg/dL

Administer 6 U of regular insulin intravenously. Recheck blood glucose in 1 hour

Severe hypoglycemia was typo.

We changed “severe hypoglycemia” to “severe hyperglycemia (> 350 mg/dL)”

2. What’s the reason for hypoglycemia? Is it due to TH or is it part of post-resuscitation syndrome? The EAG group has lower basal level of glucose compared to the standard group, If TH induces hypoglycemia, then it is not surprising that the EAG has higher incidence of hypoglycemia.

Answer) Generally, cardiac arrest patients show hyperglycemia after ROSC. Therefore, we administer insulin and avoid glucose-containing fluid. Additionally, as you pointed out, the patient undergoes therapeutic hypothermia. Thus, hypoglycemia may be because of an effect of therapeutic hypothermia, effect of treatment with insulin or non-glucose fluid, or a combined effect of both. Nevertheless, it is clear that the EAG group had a higher risk for development of hypoglycemia than the standard group since the mean glucose of EAG group was close to hypoglycemia.

The following sentences are added in the Discussion

“Although it is not obvious whether TH and/or treatment including insulin and glucose-free fluid cause hypoglycemia, it seems clear that the EAG group had a higher risk for development of hypoglycemia since it had lower mean glucose level than the standard group.”

3. The 6-month neurological outcome was evaluated by a “phone interview” rather than a neurological exam, the accuracy of the response may need to be confirmed.

Answer) The primary outcome of this study was the Cerebral Performance Category Scale which is widely used in the cardiac arrest research field because it can be measured with simple, structured questions (Longstreth WT et al. Resuscitation 2010;81:530–3.) without a practical neurologic exam. Only a board-certified emergency physician interviewed the patients or patient representatives. We have described the qualifications of the researcher who conducted the phone interview as follows:

“We assessed 6-month neurologic outcome using the cerebral performance category (CPC) scale via phone interview” was changed to “The Cerebral Performance Category (CPC) scale was assessed by a board-certified emergency physician via phone interview of patients or patient representatives and”.

4. The conclusion is very vague. The EAG group and standard group had similar neurological outcomes, though the EAG groups had higher incidence of hypothermia.

Answer) We added the following sentence in the conclusion

“Therefore, the EAG level is inappropriate as the target glucose level during post-cardiac arrest care.”

5. In 3.1, the first two paragraphs are exactly the same.

Answer) We took a mistake while organizing the manuscript according to the form of this Journal. We deleted one.

Reviewer 2 Report

Regarding the paper “Association between achievement of estimated average glucose level and 6-month neurologic outcome in comatose cardiac arrest survivors: A propensity score-matched analysis”. This paper assesses whether the achieved glucose level in relation to the HbA1C influences outcome in post cardiac arrest patients who have undergone TTM. This analysis to me appears novel and not without interest. I have the following comments:

In essence the authors suggest that the effect of achieved glucose on outcome differs based on what the patient´s “standard” glucose level is. My suggestion for studying this would be an interaction model including achieved mean glucose and HbA1C. This study appears to include patients with and without diabetes. My initial impression would be that it cannot be appropriate to include both in the same analysis. My suggestion would be to focus on those without a history of diabetes. It would be informative to have a fit plot showing the relationship between both glucose and HbA1C with the adjusted outcome. This is important also for the analysis. It may well be that hypoglycaemia and hyperglycemia are both bad and if that is the case glucose should not be included as a continuous variable. Another option could be to have a 3D diagram with achieved glucose in tertiles (x-axis), HbA1C (z-axis) and the unadjusted and adjusted outcome on the y-axis (two separate diagrams). Please have a look at Vaahersalo et al. Crit Care Med. 2014 Jun;42(6):1463-70) The paper does not include any data on how the different blood glucose levels were achieved in the two groups. If one think that the possible clinical extrapolation of these findings would be to use the HbA1C and decide on what the optimal glucose target is, then one would use insulin. So the question is, did the EAG group receive different doses of insulin and/or glucose than the other group? Or is it just that the “injury” is more severe in those who remain hyperglycaemic? The authors may want to discuss the fact that indeed in cardiac arrest patient hyperglycemia is commonly seen. We have shown it (Skrifvars MB, Pettilä V, Rosenberg PH, Castrén M. A multiple logistic regression analysis of in-hospital factors related to survival at six months in patients resuscitated from out-of-hospital ventricular fibrillation. Resuscitation. 2003 Dec;59(3):319-28) this and other have too. Adrenaline may play a role but other factors may do too. In addition as these papers have brain injury they may be more vulnerable that the regular ICU patients. Therefore, please refocus a little bit the introduction. Please provide in a supplemental table the results of the full multivariable analysis. Which factors were included and which ones were retained in the final model? Did the authors rule out co-linearity between the covariates? Why did the authors decide on a backwards stepwise approach rather than simply entering all variables in the same model?

Author Response

We appreciate you for kind review. They were very helpful in improving our manuscript and included very useful points that we had not previously recognized. After due consideration, the manuscript was revised as described below.

In essence the authors suggest that the effect of achieved glucose on outcome differs based on what the patient´s “standard” glucose level is. My suggestion for studying this would be an interaction model including achieved mean glucose and HbA1C. This study appears to include patients with and without diabetes. My initial impression would be that it cannot be appropriate to include both in the same analysis. My suggestion would be to focus on those without a history of diabetes. It would be informative to have a fit plot showing the relationship between both glucose and HbA1C with the adjusted outcome. This is important also for the analysis. It may well be that hypoglycaemia and hyperglycemia are both bad and if that is the case glucose should not be included as a continuous variable. Another option could be to have a 3D diagram with achieved glucose in tertiles (x-axis), HbA1C (z-axis) and the unadjusted and adjusted outcome on the y-axis (two separate diagrams). Please have a look at Vaahersalo et al. Crit Care Med. 2014 Jun;42(6):1463-70)

Answer) In our opinion, HbA1c is more valuable than the history of diabetes because some patients with high HbA1c did not have a history of diabetes and some patients with a history of diabetes did not have high HbA1c. It would be impossible to show a fit plot of a nominal variable with hypoglycemia because no mean glucose level represents hypoglycemia. Therefore, as per your recommendation, we developed an interaction model between mean glucose and HbA1c. We analyzed the data using a 3D diagram, as per your suggestion. This facilitated our understanding of the interaction between HbA1c and mean glucose. We divided mean glucose into tertiles and HbA1c into three groups (non-diabetic, 4.0%–5.6%; pre-diabetic, 5.7%–6.4%; diabetic, 6.5%–11.8%). We constructed the 3D diagram as described in a previous article (Vaahersalo et al. Crit Care Med. 2014 Jun;42(6):1463-70). The non-diabetic and pre-diabetic groups showed a similar pattern, but the diabetic groups showed the opposite tendency. We have described this in the Methods and Results sections. In addition, we performed univariate and multivariate analysis in the subgroups of HbA1c (HbA1c of 4.0–6.4% and HbA1c of 6.5%–11.8%) because the interaction model between HbA1c and mean glucose showed that the pattern of adjusted unfavorable neurologic outcome in HbA1c of 6.5%–11.8% was different from that of other subgroups (HbA1c of 4.0–6.4%).

We added the following sentence in the Method

“Mean glucose was divided into tertiles, and HbA1c was divided into three groups as follows: non-diabetic (4.0%–5.6%), prediabetic (5.7%–6.4%), and diabetic (6.5%–11.8%) [15].”

“Covariates which were independent predictors of neurologic outcome were included in the final model (Table S1 and Table S2). To identify the interaction between HbA1c and mean glucose, we calculated observed and adjusted neurologic outcome in the groups created according to the mean glucose tertiles and the three subgroups of HbA1c. We defined adjusted neurologic outcome as the observed unfavorable neurologic outcome in each group divided by the predicted outcome calculated from the final multivariate logistic regression model except for HbA1c. Univariate association between the glucose group and outcome and the final model adjusted for the multivariate association were tested in the subgroups of HbA1c.”

We added the following sentence in the Result.

“The observed and adjusted unfavorable neurologic outcome according to tertiles of mean glucose and subgroups of HbA1c is shown in Fig. 4.”

We added Figure 4. And legend of Figure 4 as following.

“Figure 4. Observed and adjusted neurologic outcome. (A) The observed unfavorable neurologic outcome increased with an increase in the HbA1c and mean glucose. (B) Adjusted unfavorable neurologic outcome increased in the non-diabetic (4.0%–5.6%) and prediabetic (5.7%–6.4%) groups whereas adjusted unfavorable neurologic outcome decreased in the diabetic (6.5%–11.8%) group with an increase in the mean glucose. Adjusted unfavorable neurologic outcome increased in the first and second mean glucose tertiles with an increase in the HbA1c. However, the adjusted unfavorable neurologic outcome in the third mean glucose tertile was constant irrespective of HbA1c. ”

We revised Table 4. as following.

Table 4. Odds ratios of estimated average glucose versus standard group for unfavorable neurologic outcome.

Model

Number

Odds ratio (95% confidence interval)

p

Crude

384

0.444 (0.278–0.707)

0.001

Non-diabetic and pre-diabetic

303

0.477 (0.282 – 0.804)

0.006

Diabetic

81

0.718 (0.218 – 2.361)

0.586

Adjusted

384

0.896 (0.455–1.764)

0.750

Non-diabetic and pre-diabetic

303

0.974 (0.469 – 2.022)

0.943

Diabetic

81

0.314 (0.037 – 2.697)

0.291

Crude in matched cohort

186

1.176 (0.617–2.242)

0.622

Non-diabetic and pre-diabetic

150

1.273 (0.635 – 2.551)

0.497

Diabetic

36

1.083 (0.158 – 7.435)

0.935

Adjusted in matched cohort

186

0.540 (0.143–2.034)

0.362

Non-diabetic and pre-diabetic

150

1.157 (0.448 – 2.990)

0.763

Diabetic

36

3.827 (0.111 – 131.563)

0.457

The paper does not include any data on how the different blood glucose levels were achieved in the two groups. If one think that the possible clinical extrapolation of these findings would be to use the HbA1C and decide on what the optimal glucose target is, then one would use insulin. So the question is, did the EAG group receive different doses of insulin and/or glucose than the other group? Or is it just that the “injury” is more severe in those who remain hyperglycaemic?

Answer) The EAG group and standard group were treated according to the same protocol. We have added the glucose control protocol as Table 1. Figure 2 shows the actual blood glucose level in the two groups. As shown in the figure, EAG groups generally have lower mean glucose level than the standard group. Therefore, less insulin was administered in the EAG group than the standard group, although both groups were treated according to the same protocol. Both patients with severe injury and those having high HbA1c had hyperglycemia. Moreover, HbA1c was associated with hyperglycemia and ischemic-reperfusion injury after cardiac arrest. Therefore, the interaction model that was suggested by you was an ideal representation of the data.

The authors may want to discuss the fact that indeed in cardiac arrest patient hyperglycemia is commonly seen. We have shown it (Skrifvars MB, Pettilä V, Rosenberg PH, Castrén M. A multiple logistic regression analysis of in-hospital factors related to survival at six months in patients resuscitated from out-of-hospital ventricular fibrillation. Resuscitation. 2003 Dec;59(3):319-28) this and other have too. Adrenaline may play a role but other factors may do too. In addition as these papers have brain injury they may be more vulnerable that the regular ICU patients. Therefore, please refocus a little bit the introduction.

Answer) We added the following sentences in the Introduction and we added corresponding references.

“Hyperglycemia after cardiac arrest is one of the common adverse events, and it is associated with increased mortality or unfavorable neurologic outcome [3-5]. Counter-regulatory hormones such as catecholamines are known to cause stress-induced hyperglycemia in critical illness [6].”

Please provide in a supplemental table the results of the full multivariable analysis. Which factors were included and which ones were retained in the final model? Did the authors rule out co-linearity between the covariates? Why did the authors decide on a backwards stepwise approach rather than simply entering all variables in the same model?

Answer) We have revised the supplemental tables (Supplemental Table 1 and Supplemental Table 2) and indicated which factors were included in the final model. To avoid overfitting, we selected covariates that met the requirement of “one in ten” rule. Therefore, we followed a backward, stepwise approach.

We performed collinearity test. The following sentence was added in the Methods

“The collinearity between variables was assessed before modeling.”

Supplemental Table S1. Univariate and multivariate logistic regression analyses for unfavorable neurologic outcome in entire cohort

Crude OR (95% CI)

p

Adjusted OR (95% CI)

p

Age, years

1.046 (1.030–1.062)

<0.001

1.042 (1.019–1.065)*

<0.001

Male

0.559 (0.352–0.886)

0.013

0.579 (0.297–1.127)

0.108

Coronary artery disease

0.866 (0.499–1.501)

0.607

NA

Congestive heart failure

1.173 (0.570–2.416)

0.665

NA

Hypertension

1.556 (1.021–2.403)

0.040

0.739 (0.363–1.505)

0.404

Diabetes

3.157 (1.870–5.329)

<0.001

1.054 (0.443–2.511)

0.905

Pulmonary disease

2.888 (0.821–10.158)

0.098

1.843 (0.323–10.515)

0.491

Renal impairment

3.116 (1.414–6.868)

0.005

1.346 (0.460–3.941)

0.587

Cerebrovascular accident

1.636 (0.674–3.974)

0.277

NA

Hepatic disease

2.810 (0.325–24.299)

0.348

NA

Body mass index, kg/m2

0.976 (0.925–1.030)

0.383

NA

OHCA

0.908 (0.517–1.593)

0.735

NA

Witnessed

0.342 (0.197–0.596)

<0.001

0.593 (0.268–1.312)

0.197

Bystander CPR

0.628 (0.404–0.979)

0.040

1.049 (0.523–2.104)

0.892

Shockable rhythm

0.107 (0.066–0.174)

<0.001

0.246 (0.114–0.531)*

<0.001

Cardiac etiology

0.179 (0.109–0.292)

<0.001

0.268 (0.126–0.567)*

0.001

Adrenaline, mg

1.207 (1.107–1.316)

<0.001

0.931 (0.831–1.042)

0.211

Time to ROSC, min

1.042 (1.026–1.058)

<0.001

1.062 (1.038–1.087)*

<0.001

HgA1c, %

1.544 (1.224–1.948)

<0.001

1.411 (1.049–1.897)*

0.023

Hemoglobin, mg/dL

0.762 (0.694–0.837)

<0.001

0.923 (0.796–1.070)

0.289

Lactate, mmol/L

1.124 (1.062–1.189)

<0.001

1.100 (1.012–1.195)*

0.025

Glucose, mg/dL

1.003 (1.001–1.005)

0.004

1.001 (0.997–1.005)

0.598

PaO2, mmHg

1.003 (1.001–1.005)

0.013

1.005 (1.001–1.008)*

0.010

PaCO2, mmHg

1.013 (0.997–1.029)

0.109

1.018 (0.993–1.043)

0.155

Glasgow Coma Scale

0.587 (0.493–0.698)

<0.001

0.737 (0.595–0.913)*

0.005

SOFA score

1.215 (1.134–1.302)

<0.001

1.091 (0.986–1.206)

0.090

Time from ROSC to TH, min

1.002 (1.000–1.004)

0.046

1.003 (1.000–1.006)*

0.029

Induction duration, h

0.737 (0.656–0.830)

<0.001

0.843 (0.716–0.994)*

0.042

Rewarming duration, h

1.107 (1.026–1.195)

0.009

1.028 (0.920–1.148)

0.631

OR, odds ratio; CI, confidence interval; NA, not applicable; OHCA, out-of-hospital cardiac arrest; CPR, cardiopulmonary resuscitation; ROSC, restoration of spontaneous circulation; HbA1c, glycated hemoglobin; PaO2, partial pressure of oxygen; PaCO2, partial pressure of carbon dioxide; SOFA, sequential organ failure assessment; TH therapeutic hypothermia.

* Variables included in the final model.

Supplemental Table S2. Univariate and multivariate logistic regression analyses for unfavorable neurologic outcome in matched cohort

Crude OR (95% CI)

p

Adjusted OR (95% CI)

p

Age, years

1.029 (1.007–1.051)

0.010

1.017 (0.985–1.050)

0.295

Male

0.563 (0.274–1.157)

0.118

0.870 (0.297–2.547)

0.800

Coronary artery disease

0.812 (0.343–1.923)

0.635

NA

Congestive heart failure

3.062 (0.675–13.896)

0.147

4.633 (0.741–28.968)

0.101

Hypertension

0.760 (0.398–1.451)

0.405

NA

Diabetes

1.852 (0.848–4.045)

0.122

0.717 (0.223–2.304)

0.779

Pulmonary disease

1.750 (0.365–8.389)

0.484

NA

Renal impairment

1.851 (0.509–6.732)

0.350

NA

Cerebrovascular accident

1.561 (0.422–5.776)

0.505

NA

Hepatic disease

NA

NA

Body mass index, kg/m2

1.008 (0.932–1.092)

0.837

NA

OHCA

1.188 (0.521–2.708)

0.682

NA

Witnessed

0.154 90.052–0.454)

0.001

0.364 (0.093–1.435)

0.149

Bystander CPR

0.324 (0.150–0.700)

0.004

0.306 (0.102–0.912)*

0.034

Shockable rhythm

0.087 (0.041–0.184)

<0.001

0.056 (0.020–0.159)*

<0.001

Cardiac etiology

0.189 (0.089–0.400)

<0.001

0.560 (0.163–1.925)

0.358

Adrenaline, mg

1.174 (1.024–1.346)

0.021

0.843 (0.696–1.020)

0.079

Time to ROSC, min

1.037 (1.014–1.061)

0.001

1.080 (1.036–1.126)*

<0.001

HgA1c, %

1.385 (0.935–2.051)

0.104

1.264 (0.757–2.109)

0.370

Hemoglobin, mg/dL

0.803 (0.700–0.921)

0.002

0.803 (0.658–0.980)*

0.031

Lactate, mmol/L

1.077 (0.991–1.171)

0.080

1.066 (0.951–1.194)

0.270

Glucose, mg/dL

0.998 (0.995–1.002)

0.404

NA

PaO2, mmHg

1.004 (1.000–1.008)

0.029

1.007 (1.001–1.013)*

0.017

PaCO2, mmHg

1.013 (0.990–1.036)

0.278

NA

Glasgow Coma Scale

0.618 (0.477–0.802)

<0.001

0.891 (0.617–1.287)

0.539

SOFA score

1.141 (1.030–1.264)

0.012

1.205 (1.016–1.430)*

0.033

Time from ROSC to TH, min

1.002 (0.999–1.004)

0.223

NA

Induction duration, h

0.777 (0.639–0.945)

0.011

1.039 (0.738–1.464)

0.825

Rewarming duration, h

1.012 (0.890–1.1520

0.852

NA

OR, odds ratio; CI, confidence interval; NA, not applicable; OHCA, out-of-hospital cardiac arrest; CPR, cardiopulmonary resuscitation; ROSC, restoration of spontaneous circulation; HbA1c, glycated hemoglobin; PaO2, partial pressure of oxygen; PaCO2, partial pressure of carbon dioxide; SOFA, sequential organ failure assessment; TH therapeutic hypothermia. * Variables included in final model.

Round 2

Reviewer 1 Report

All my questions were answered. 

Author Response

We checked again overall to correct grammar and spelling. The revised part has been marked.

Thank you for kind review.

Reviewer 2 Report

I have read with interest this much improved version. I think this is an important paper that will be of interest to the readers of JCM

I have one suggestion. The results now suggest that perhaps the group that benefits the least from more "intensive insulin" therapy are the ones with a high HbA1C. This is in fact a much studied matter and there are studies ongoing on this. Please have, for example a look at the study Luethi et al. Acta Anaest Scand 2019 63 (6) 761-768. I think this is something worth discussing in the paper.  

Author Response

Thank you again for your kind review and suggestion.

I have one suggestion. The results now suggest that perhaps the group that benefits the least from more "intensive insulin" therapy are the ones with a high HbA1C. This is in fact a much studied matter and there are studies ongoing on this. Please have, for example a look at the study Luethi et al. Acta Anaest Scand 2019 63 (6) 761-768. I think this is something worth discussing in the paper.

Answer) Thanks to the article that you recommend, we have become aware of research trends on glucose control in the chronic hyperglycemic status and we found more articles those presented that liberal glucose control is better in critically ill patient with chronic hyperglycemia. Therefore, we changed the following sentence in the Discussion

“However, the diabetic subgroup (HbA1c >6.5%) had a small sample size in the present study. Further study is warranted to investigate the association between hyperglycemia and neurologic outcome based on the premorbid glycemic status in a large group of patients.”

To

Previous studies demonstrated that liberal (180–252 mg/dL) glucose control was not associated with increased mortality or adverse events, whereas reduced hypoglycemia and glucose variability than conventional (108–180 mg/dL) glucose control in critically ill patients with diabetes or patients with chronic hyperglycemia (HbA1c ≥ 7%) [21-23]. Those previous research support that HbA1c based glucose control might be better to avoid potential risk of adverse events in cardiac arrest survivors with chronic hyperglycemia or poorly controlled diabetes.

We also analyzed the association between hypoglycemia and glucose control groups in subgroups of nondiabetic and prediabetic patients and in diabetic subgroup. We found there is no association between diabetic group and hypoglycemia. We added the following sentences in the results

EAG group was associated with increased hypoglycemia only in subgroup of nondiabetic and prediabetic patients (EAG, 31/210 versus standard 6/93; p = 0.010), whereas was not associated with hypoglycemia in diabetic subgroup (EAG, 9/38 versus standard 5/43; p = 0.152).”

EAG group was associated with increased hypoglycemia only in subgroup of nondiabetic and prediabetic patients (EAG, 18/78 versus standard 3/72; p = 0.001), whereas was not associated with hypoglycemia in diabetic subgroup (EAG, 3/15 versus standard 3/21; p = 0.677).”

We changed the following sentences in the Discussion

“The association between hypoglycemia and unfavorable neurologic outcome is unclear in the present work. However, it seems reasonable to recommend the higher target glucose level over the EAG, to avoid potential harm owing to hypoglycemia in cardiac arrest survivors.”

To

However, diabetic subgroup with the HbA1c ≥6.5% was not associated with hypoglycemia, unlike the non-diabetic and pre-diabetic subgroups in the present study. Previous studies demonstrated that liberal glucose control reduced hypoglycemia in critically ill patients with HbA1c ≥7% [22, 23]. Thus, even if the association between hypoglycemia and unfavorable neurologic outcome is unclear in the present study, it might be acceptable to recommend the EAG based glucose control in cardiac arrest survivors with chronic hyperglycemic status to avoid potential harm owing to hypoglycemia.”

We revised the conclusion as following

“Achievement of EAG levels during TH in comatose cardiac arrest survivors was not associated with 6-month neurologic outcome or serum NSE in our propensity score-matched cohort, whereas was associated with a high prevalence of hypoglycemia. However, the associations seem to be invalid in diabetic subgroup. Therefore, further randomized clinical trials are warranted to identify the optimal glucose range based on the chronic glycemic status after ROSC in cardiac arrest survivors.

Thanks